# Acid Number Prediction Model of Lubricating Oil Based on Mid-Infrared Spectroscopy

**Fanhao Zhou** [1,2,3], **Kun Yang** [1,2,*], **Dayang Li** [1,2] **and Xinfa Shi** [4]

1 Reliability Engineering Institute, School of Transportation and Logistics Engineering, Wuhan University of Technology, Wuhan 430063, China

2 National Engineering Research Center for Water Transport Safety, Wuhan University of Technology, Wuhan 430063, China

3 School of Ship, Ocean, Energy and Power Engineering, Wuhan University of Technology, Wuhan 430063, China

4 Guangzhou Institute of Mechanical Science Co., Ltd., Guangzhou 510700, China

* Correspondence: kyang@whut.edu.cn

**Abstract:** The monitoring and replacement of lubricating oil has an important impact on mechanical equipment. In this study, based on the infrared spectroscopy monitoring method, an acid value index prediction model is established. The support vector machine regression method is used to quantitatively analyze the acid number of the oil sample, which verifies the stability and predictive ability of the quantitative prediction model, and we provide a theoretical basis and practical examples for the online monitoring of oil indicators. In addition, a support vector machine regression model is established by observing the changing law of the spectral absorption peak and selecting the dominant wavelength, and it is compared against the prediction algorithm of the long- and short-term memory network. By comparing the deviation relationship between the predicted value and the real chemical value, the feasibility of the infrared spectroscopy prediction model is verified. The experimental results show that the correlation coefficient between the predicted value of the model and the actual measured value reaches 0.98. This proves that the prediction effect of the prediction model based on the infrared spectrum data and the support vector machine regression method is better than that of the long- and short-term memory network trend prediction model, and the predicted results are reliable.

**Keywords:** lubricating oil; oil monitoring; mid-infrared spectroscopy; acid number; support vector regression

## 1. Introduction

Lubricating oil, which provides the functions of lubrication, cleaning, cooling, sealing, anti-wear, anti-rust, anti-corrosion, etc., has a profound effect on the operating conditions of various devices. Lubricating oil comes into direct contact with metal during use and is affected by various environmental factors. Due to the oxidation reaction during use, the physical and chemical properties are changed and corresponding metamorphic substances are derived. The substances will be eroded by sand and dust, which results in large amounts of harmful substances in the oil [1]. However, once the impurities and oxides in the oil exceed the standard, the production of an oil film during use becomes difficult. As a result, the anti-corrosion and lubricating properties of the oil are significantly weakened. This leads to serious friction and wear of the equipment, and even serious failures [2]. Thus, oil quality has a profound impact on the service life of the equipment. Furthermore, the use of low-quality oil will make it difficult for the equipment to maintain normal operating conditions. This will result in some or all of the functions being unrealized, leading to reduced operation efficiency, increased instability of the system operation, and increased noise pollution. Simultaneously, it will cause a risk of high wear of the moving parts in the device, high temperature of the supercharger, and even engine damage [3]. According

to statistical data, 80% of mechanical device operation failures are due to increased wear caused by substandard oil quality, which in turn causes failure and damage to the mechanical equipment [4]. Conclusively, the quality of the oil has a decisive influence on the service cycle and operating efficiency of the equipment [5], and, hence, it is necessary to monitor the quality of lubricating oil.

Infrared spectroscopy can measure important indicators such as the oxidation value, acid-base value, and moisture in oil. The monitoring of oil quality can be achieved by monitoring these indicators in the oil. The benefits of this method are its ease of operation, faster detection speed as compared to other methods, and lack of direct contact with the oil. The main detection methods based on infrared spectroscopy include the neutralization reaction [6,7], nitrate sulfide value and oxidation value combined with total acid number [8,9], partial least square regression method [10,11], neural network method [12], and other methods [13,14]. In recent years, Lingfei Shi [15] used the method of combining least squares and support vector machine to establish a lubricating oil acid number model, and compared it with the accuracy of prediction based on the radial basis function model. Yanjun Zhang [16] and others established a rapid quantitative detection model of acid content based on artificial bee colony support vector regression and Raman spectroscopy. This method demonstrated higher prediction accuracy. Juxiang Wang et al. [17] used the spectral variables after feature selection to establish an acid number analysis model based on the correlation vector machine algorithm, and verified the accuracy. FRVD Voortd et al. [18] used infrared spectroscopy to determine the acid number and number of bases in lubricating oil, and eliminated the matrix effect through a signal transduction method combining chemometrics and differential spectroscopy. R Chakravarthy [19] and others used mid-infrared spectroscopy to determine the naphthenic acid number in petroleum crude oil and its fractions. Ran Zhiyong et al. established the relationship between near-infrared spectroscopy and oil use times, and used forward interval partial least squares (FiPLS) and backward interval partial least squares (BiPLS) to screen feature intervals [20].

The above research illustrates the feasibility and wide application of infrared spectroscopy in the detection of the lubricating oil acid number, and all experiments verified the accuracy of infrared spectroscopy monitoring. It is not difficult to find that algorithms are increasingly used in infrared spectroscopy From data acquisition methods to chemometric methods used in regression modeling, continuous improvements are being made, and the scope of application is gradually expanding. The stability of the model is gradually improving, and the deviation of prediction is gradually shrinking. However, most of the known research methods involve the prediction of the existence of oil samples with different oxidation times. Therefore, a prediction method for the production of oxidized lubricating oil at a future time is needed. In this study, a prediction method based on support vector machine is proposed. Given its high accuracy, the use of infrared spectroscopy for oil quality prediction research is able to achieve results closer to the actual state.

## 2. Materials and Methods

Using the infrared spectroscopy detection method, the correlation between the predicted spectral data and the data results measured in the laboratory environment can be analyzed, which is conducive to the rapid determination of oil products. The regression method of support vector machine was used to construct a corresponding quantitative model to analyze the acid number of oil, which was then used to verify the accuracy of the acid number compared to that measured by infrared spectroscopy.

### 2.1. Collection of Experimental Data

2.1.1. Infrared Spectrum Collection

The lubricating oil sample used in the experiment was L-TSA46 steam turbine base oil. First, 100 mL of L-TSA46 steam turbine oil in a round-bottom flask was weighed, and then a digital heater was used to heat the base oil at the same temperature to simulate oxidation. In total, 37 samples for different dates were collected. A Nicolet Avatar 360 FT-IR Fourier

Transform Infrared Spectrometer (Thermo Nicolet Corporation, Madison, WI, USA) was used to analyze the oil sample, and measure and record its spectral data. The infrared spectrum of the sample was collected at a room temperature of 25 °C, with the atmospheric environment as the background. The spectrum collection range was 4000~400 cm$^{-1}$, the resolution was 4 cm$^{-1}$, and the number of scans was 32.

2.1.2. Acid Number Data Determination

The acid number, which refers to the amount of alkali consumed in the process of neutralizing the acidic components in 1 g of oil, and whose unit is mgKOH·g$^{-1}$, is one of the most important research parameters of oil [21]. In this research work, the acid number detection method for petroleum products was used to test the acid number of the sample. Potassium hydroxide solution was then used for titration detection. Combined with the amount of the final titration solution, the acid number of the sample was calculated. The testing equipment used was the Mettler G20S automatic titrator (METTLER TOLEDO, Zurich, Switzerland).

According to the standard measurement method, the acid number measurement results for the 37 samples are listed in Table 1.

**Table 1.** Acid numbers of lubricating oil samples.

| Number | Acid Number mgKOH/g | Number | Acid Number mgKOH/g | Number | Acid Number mgKOH/g |
|---|---|---|---|---|---|
| 1 | 0.032 | 14 | 0.043 | 27 | 0.043 |
| 2 | 0.035 | 15 | 0.046 | 28 | 0.046 |
| 3 | 0.036 | 16 | 0.036 | 29 | 0.048 |
| 4 | 0.034 | 17 | 0.038 | 30 | 0.047 |
| 5 | 0.036 | 18 | 0.04 | 31 | 0.049 |
| 6 | 0.038 | 19 | 0.038 | 32 | 0.046 |
| 7 | 0.033 | 20 | 0.038 | 33 | 0.048 |
| 8 | 0.037 | 21 | 0.041 | 34 | 0.049 |
| 9 | 0.042 | 22 | 0.045 | 35 | 0.05 |
| 10 | 0.039 | 23 | 0.042 | 36 | 0.051 |
| 11 | 0.035 | 24 | 0.047 | 37 | 0.053 |
| 12 | 0.045 | 25 | 0.04 | - | - |
| 13 | 0.042 | 26 | 0.042 | - | - |

*2.2. Data Processing*

2.2.1. Choice of Dominant Bands

Under the influence of oxidation, chain alkyl radicals reacted with oxygen early to generate oxygen radicals. It was then formed from another hydrocarbon that extracts hydrogen from hydrogen peroxide and another free radical. With the production and accumulation of hydrogen peroxide, the final oxidation process of the oil was terminated. The production of carboxylic acids increased the acidity of the oil. The absorption peaks of the functional groups of the above substances were generally distributed across the entire spectrum. Under the influence of hydroxyl absorption, the absorption bands of carboxylic acid substances appeared to be diffuse, and the absorption peak width ranges were from 3770 to 3100 cm$^{-1}$. However, this segment overlapped with the absorption peak of moisture and hence could not be used as a basis for the quantitative analysis of the acid number. Moreover, 1300~1000 cm$^{-1}$ was the C-O single bond stretching vibration peak of the oxygen-containing compound. However, in addition to carboxylic acids, the acidic substances in the oil also included a variety of organic acids and acidic additives, or other unknown substances that could react with the -OH bonds. Therefore, this section could not be selected for quantitative analysis. Referencing the ASTM E 1421-99 (2015) e1 standard, 1800~1600 cm$^{-1}$ is an important indication range for determining the oxidation value of oil by infrared spectroscopy. The absorption peak in this waveband displayed obvious

changes. Therefore, the band of 1800~1600 cm$^{-1}$ could be used to characterize the acid number. Figure 1 presents the infrared spectra of some oil samples.

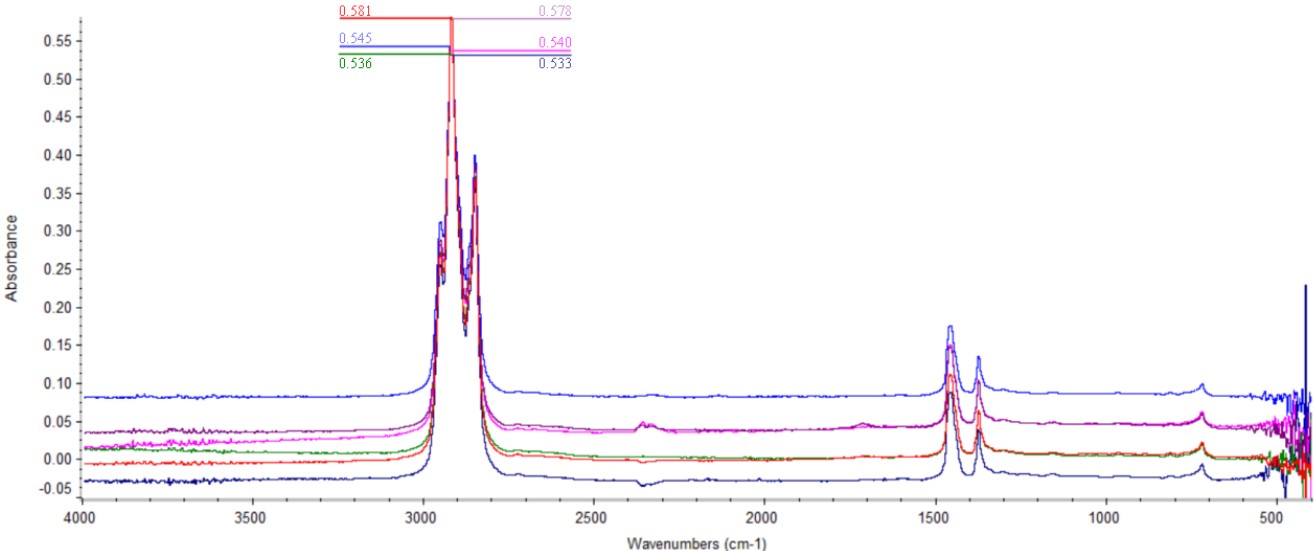

**Figure 1.** Infrared spectra of lubricating oil samples.

### 2.2.2. Selection of Spectral Preprocessing Method

In the measurement link, the infrared spectrum data will inevitably produce errors, so the infrared spectrum information needs to be preprocessed [22]. The value of preprocessing work is to eliminate irrelevant data information in the detection information and reduce information noise, which effectively reduces the influence of irrelevant elements on the spectrum. As a result, the reliability of the model and the accuracy of the results are improved. The maximum and minimum acid numbers of the lubricating oil obtained by the test were 0.053 mgKOH·g$^{-1}$ and 0.032 mgKOH·g$^{-1}$, respectively, which did not exceed the standard range, and the average value was 0.042 mgKOH· g$^{-1}$.

### 2.2.3. Sample Division

In this experiment, the Kennard–Stone method was used to divide the 37 samples of lubricants into a training set and a validation set at a ratio of 4:1, i.e., 30 samples were used as the training set and 7 samples were used as the validation set for training. The details are shown in Table 2.

**Table 2.** Sample division.

| Sample Set | Number | Acid Number Range mgKOH/g | Average Value mgKOH/g | Standard Deviation mgKOH/g |
|---|---|---|---|---|
| Training set | 30 | 0.032~0.048 | 0.040 | 0.0045 |
| Validation set | 7 | 0.046~0.053 | 0.049 | 0.0022 |

### *2.3. Modeling and Preprocessing*

### 2.3.1. Modeling

The LIBSVM toolbox was selected in MATLAB 2020b (MathWorks, Natick, MA, USA) for modeling. It is a toolbox with SVM pattern recognition and regression functions, developed by Zhiren Lin scholars, which can effectively solve the open-source support vector machine classification and regression problems [23].

### 2.3.2. Normalization

In order to improve the modeling efficiency and computing speed, this research used normalization to preprocess the lubricant sample data. Because of the differences in the light absorption capacity of different substances, the corresponding data on the acid number of lubricating oils were also different. Therefore, when training samples with different data ranges were used, data convergence would slow down during the modeling process, affecting the modeling work. In the experiment, the mapminmax function of MATLAB was used to realize the normalization of the interval [−1,1].

### 2.3.3. Optimization of Parameters

The penalty function $c$ and the kernel function $g$ are key functions in the model building process. The generalization ability of the model is deeply affected by the penalty factor. At present, the commonly used penalty functions include root mean square error and mean square error.

The root mean square error (*RMSE*) calculation formula is as follows:

$$RMSE = \sqrt{\frac{1}{T}\sum_{t=1}^{T}(y_t - \hat{y}_t)^2} \tag{1}$$

The mean square error (*MSE*) calculation formula is as follows:

$$MSE = \frac{1}{T}\sum_{t=1}^{T}(y_t - \hat{y}_t)^2 \tag{2}$$

In the above formula, $y_t$ —predicted value; $\hat{y}_t$—measured value; $T$—the number of predicted values.

In the process of this research, the penalty parameter $c$ was searched within a certain range, and at the same time, a limited condition was added. Therefore, the parameter combination with the smaller penalty parameter $c$ was preferentially selected. In addition, for the regression prediction model, the degree of fit ($R^2$) was used to evaluate the accuracy of the model.

$$R^2 = 1 - \frac{\sum_{t=1}^{T}(y_t - \hat{y}_t)^2}{\sum_{t=1}^{T}(y_t - \overline{y_t})^2} \tag{3}$$

The research results show that the kernel function $g$ of the LIBSVM toolbox satisfies the conditions of the kernel function theorem. However, in the actual application process, the alternative kernel functions are RBF, polynomial, linear, and sigmoid, with four types of kernel functions in order, which can effectively verify the effect of the kernel function on the performance of the regression model in this experiment. The research results showed that when RBF was selected as the kernel function, the generalization ability of the model achieved the best outcome. Therefore, in this research work, the RBF kernel function was selected in terms of the kernel function. When using this function, due to its own parameter gamma, i.e., the parameter $g$, there is a negative correlation between the size of $g$ and the number of support vectors. However, the number of the latter has a profound impact on the speed of training and prediction of the entire SVR model. The gamma calculation formula in the RBF kernel function is shown below.

$$k(x,z) = \exp\left(-\frac{d(x,z)^2}{2\sigma^2}\right) = \exp\left(-gamma \cdot d(x,z)^2\right) \tag{4}$$

Based on the physical meaning of $g$, it can be inferred that when the gamma is set to be too large, the predictive ability will be weakened. It can only predict the vicinity of the sample and is unable to predict the unknown sample, which causes training problems.

However, if the value is set too low, it will cause the smoothing effect to be too significant, and the accuracy of the validation set and training set will be weakened.

## 3. Results and Discussion

On the basis of the above research, the method of support vector machine regression was used in MATLAB and the method of long short-term memory (LSTM) [24] trend prediction was used in Python to establish a quantitative prediction model for the lubricating oil acid number. In the modeling process, the hyperparameters of the model were determined by the grid search method. The hyperparameters of the LSTM trend prediction model are shown in Table 3. Substituting the infrared spectrum data and acid number data into the model, the prediction results of the training set and prediction set were obtained, as shown in Figures 2 and 3.

**Table 3.** LSTM model hyperparameters.

| Hyperparameter Name | Value |
| --- | --- |
| Dropout | 0.02 |
| Epochs | 100 |
| Batch_size | 2 |
| Activation function | Tanh |
| Optimizer | Adagrad |
| Number of neural network layers | 3 |
| Number of hidden layer neurons | 64 |
| Loss function | *MSE* |

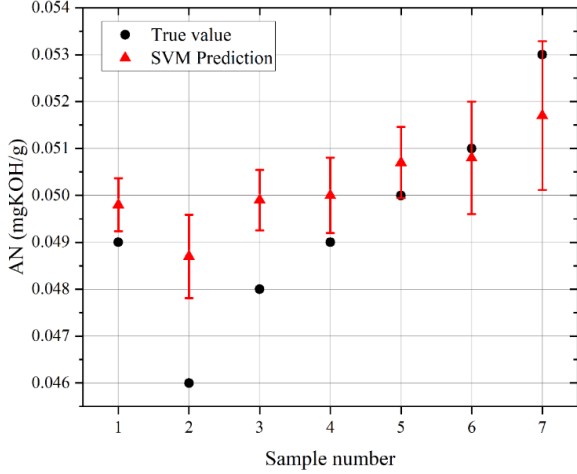

**Figure 2.** SVM training set prediction results.

Using the results in Figures 2 and 3, the errors and relative errors between the measured and predicted values in the SVM model and LSTM model verification set could be calculated, respectively, as shown in Tables 4 and 5.

**Table 4.** Error of SVM measured value and predicted value.

| Index | Measured Value mgKOH/g | Predictive Value mgKOH/g | Deviation mgKOH/g | Relative Error % |
| --- | --- | --- | --- | --- |
| 1 | 0.049 | 0.04991 | 0.00091 | 1.86 |
| 2 | 0.046 | 0.04861 | 0.00261 | 5.67 |
| 3 | 0.048 | 0.04988 | 0.00188 | 3.92 |
| 4 | 0.049 | 0.05036 | 0.00136 | 2.78 |
| 5 | 0.050 | 0.05075 | 0.00075 | 1.50 |
| 6 | 0.051 | 0.05105 | 0.00005 | 0.10 |
| 7 | 0.053 | 0.05168 | −0.00132 | 2.49 |

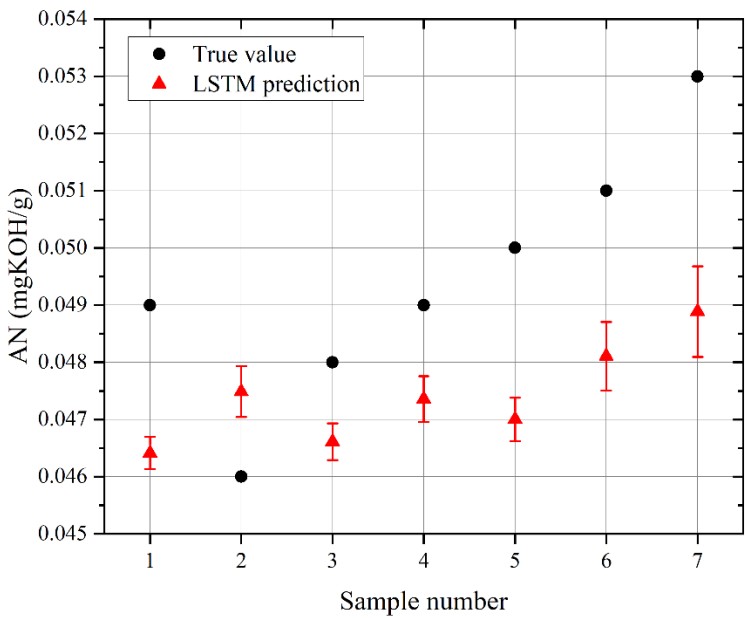

**Figure 3.** LSTM training set prediction results.

**Table 5.** Error of LSTM measured value and predicted value.

| Index | Measured Value mgKOH/g | Predictive Value mgKOH/g | Deviation mgKOH/g | Relative Error % |
|---|---|---|---|---|
| 1 | 0.049 | 0.04669 | 0.00231 | 4.72 |
| 2 | 0.046 | 0.04788 | −0.00188 | 4.10 |
| 3 | 0.048 | 0.04693 | 0.00107 | 2.23 |
| 4 | 0.049 | 0.04769 | 0.00131 | 2.66 |
| 5 | 0.050 | 0.04736 | 0.00264 | 5.28 |
| 6 | 0.051 | 0.04868 | 0.00232 | 4.56 |
| 7 | 0.053 | 0.04964 | 0.00336 | 6.33 |

The results obtained from Formula (1) are shown in Table 6.

**Table 6.** *RMSE* value of the model.

| Model | SVM | LSTM |
|---|---|---|
| *RMSE* | 0.00147994 | 0.00224797 |

It is observed that the relative error of the SVM model varies within the range of 0.004% to 0.016%, which indicates that the model based on the SVM algorithm has a small relative error and a better prediction effect. At the same time, the *RMSE* of the SVM model is smaller than that of the LSTM model, which indicates that the regression prediction effect of the SVM model is better than that of the LSTM trend prediction model. The coefficients of determination $R^2$ of the acid number of the training set and the verification set and the related infrared spectrum data are 0.959 and 0.960, respectively. This shows that the accuracy rate of the lubricating oil acid value calculated by infrared spectrum data is 96%. Therefore, the correlation coefficient $R$ can also be calculated to be 0.979 and 0.980, respectively. According to the corrected standard deviation (SEC) and predicted standard deviation (SEP) formulas, SEC and SEP can be calculated as 0.0012 and 0.0027, respectively. The small value indicates that the support vector regression model has strong predictive ability and a good fitting effect. It shows that the infrared spectrum data of the lubricating oil sample have a significant correlation with its acid number.

Through the above analysis, it can be deduced that support vector regression modeling can predict the acid number of lubricating oil well within the standard, with a small prediction error, and the prediction effect is better than that of the LSTM trend prediction model. The study confirmed that it is very feasible to determine the acid number of lubricating oil samples through Fourier transform mid-infrared spectroscopy data.

## 4. Conclusions

In this paper, the relationship between the acid number of lubricating oil and infrared spectroscopy detection technology was studied, the influence of changes in the physical and chemical indicators of lubricating oil was analyzed, and the principles, advantages, and disadvantages of infrared spectroscopy detection technology were analyzed. The data analysis methods of infrared spectroscopy were studied, including the preprocessing method, the selection of the characteristic wavelength, the analysis of commonly used quantitative modeling algorithms, and the support vector machine regression model. Finally, the infrared spectrum of the acid number of lubricating oil was quantitatively analyzed. By comparison with the trend prediction model of the lubricating oil acid number based on the LSTM algorithm, it is found that the stability and accuracy of the model built by the support vector machine regression method is better, and the correlation coefficient of the verification set is 0.98. Therefore, acid value prediction based on mid-infrared spectroscopy is a feasible method to evaluate the period for oil replacement. This has guiding significance for the protection of mechanical components.

In this experiment, infrared spectroscopy was selected as the monitoring method of oil quality because of its strong accuracy. The acid number, as an indicator for judging the degree of oil deterioration, can be obtained by monitoring acid-containing groups in oil by infrared spectroscopy. Since the oxidation products of lubricating oil have clear infrared spectral characteristics, the correlation between functional groups and oil quality is more reliable. In addition, its analysis speed is fast. However, for oil detection, infrared spectroscopy detection technology still has a certain scope of application. It can only characterize the molecular structure and functional groups, and is unable to reflect the indicator status of metal particles, atoms, and dissolved ions. Therefore, there may be slight errors in the monitoring of highly contaminated oils.

**Author Contributions:** Conceptualization, F.Z., K.Y. and D.L.; methodology, F.Z.; software, D.L.; validation, F.Z., K.Y. and D.L.; formal analysis, F.Z.; investigation, F.Z.; resources, F.Z.; data curation, F.Z.; writing—original draft preparation, F.Z.; writing—review and editing, F.Z., K.Y. and D.L.; visualization, F.Z., K.Y. and D.L.; supervision, K.Y.; project administration, K.Y.; funding acquisition, X.S. All authors have read and agreed to the published version of the manuscript.

**Funding:** National Natural Science Foundation of China (NSFC) (52071241), the Opening Project of National and Local Joint Engineering Research Center for Industrial Friction and Lubrication Technology (2021-GD-0004).

**Institutional Review Board Statement:** Not applicable.

**Informed Consent Statement:** Not applicable.

**Conflicts of Interest:** There are no relevant financial or non-financial competing interest to report. The authors declare no conflict of interest.

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
