# Peer review of "Acid Number Prediction Model of Lubricating Oil Based on Mid-Infrared Spectroscopy"

_lubricants, doi:10.3390/lubricants10090205_

Round 1

Reviewer 1 Report

You did much experimental work for this paper and present an interesting work.  From my point of view it would be improve the paper and would be interesting to present some more international literature.

Author Response

Point 1: Reviewer thinks that reference relevance can be improved.

Response 1: I increased the relevance of the references and selected references with more recent years.

Point 2: Reviewer thinks the research design could be improved.

Response 2: In this paper, the deterioration degree of oil products is predicted on the basis of spectral monitoring. I improved the prediction based on the support vector machine method.

Reviewer 2 Report

Lubricating oil acid number model based on infrared spectroscopy:

In the introduction section, could authors expressed applications or usage of the mentioned lubricating oils? Also, Infrared spectroscopy is mentioned as main characterization/evaluation of lubricant.

In figure 2, authors consider we could have a data on each of the demonstrated peaks? probably as a higlight...

I recommend that figure 3 must be prepared from the obtained data and not a screen shot from the computer. Please be considered with the regulatory standard from the journal about figures-charts.

Please include erro bars (standard deviation) in Figure 3.2

I would recommend the authors to be more detailed in the explanation for advantages and disadvantages of IR spectroscopy detection technology analyzed in this work.

What is the main application of this research? value added?

I highly recommend the search and use of more recent references (>2020)

Reviewer 3 Report

The manuscripts reports the use of Machine Learning regression methods (Support Vector Machine; and short-term memory network) to predict the acid number of a lubricating oil using data on response bands of functional groups and bands with obvious changes in the absorption peaks as experimentally obtained from infrared spectroscopy analysis and chemometrics. Overall, the topic fits the scope of MDPI Lubricants and could be interesting in the context of online monitoring of oil indicators. However, I have some doubts about the novelty or significance of the reported study and results. Just fitting ML models to a training data set, where neither the methods of data generation nor the ML approaches are novel, appears to me to be too little for a scientific publication. The authors would still need to verify their prediction with independently generated experimental data. Also, the usability of the predictive model is not entirely clear to the reviewer, as it seems to be very specific to the problem or oil formulation at hand. In addition, there is a lack of discussion or classification of the results in the context of the literature.

Apart from this severe aspect, I have various other comments:

1. The title of the paper should be improved and not contain "Research of...". It is pretty obvious that a research paper presents research results. Make it more clear what the reader might expect.

2. The abstract is too extensive.

3. The introduction is too brief, which is also reflected in the small number of also mostly older cited papers. The research gap remains unclear. What is the novelty of this study compared to published literature? The usage of machine learning or artificial intelligence methods in the context of tribology (lubricant formulation, condition monitoring etc.) is mostly ignored.

4. Section 2 should be renamed to "Materials and Methods" and not only include information on the experiments but also the employed regression methods. The "results and discussion" section should then cover results and discussion only...

5. Table 2.1 should be re-arranged as it can be confusing.

6. Clearly refer to the tables and figures in the text, e.g. "... are listed in table 1", and avoid expressions like "are listed below". Also,tables and figures should be numbered consecutively (Table 1, Table 2 etc. instead of Table 2.1).

7. If possible, provide references for the employed methods if they are rather specific (e.g. Kennard-Stone method).

8. The quality of the figures is poor and the appearance very inconsistent. This makes it difficult to read.

Overall, I therefore recommend rejection of the manuscript.

Round 2

Reviewer 2 Report

Good work.

Thanks to the authors for accepting and applying the recommended modifications and improvements to their manuscript.

Accept as it is.

Reviewer 3 Report

The authors have revised their manuscript, however have not highlighted the changes in the file. Moreover, I still have doubts about the novelty and significance of the reported study and believe it is too little for a scientific publication. Therefore, I stick with my recommendation of rejection.